# Companion Animals—An Overlooked and Misdiagnosed Reservoir of Carbapenem Resistance

**DOI:** 10.3390/antibiotics11040533

**Published:** 2022-04-17

**Authors:** Joana Moreira da Silva, Juliana Menezes, Cátia Marques, Constança Ferreira Pomba

**Affiliations:** 1Centre for Interdisciplinary Research in Animal Health (CIISA), Faculty of Veterinary Medicine, University of Lisbon, 1300-477 Lisbon, Portugal; jmsilva@fmv.ulisboa.pt (J.M.d.S.); julianamenezes@fmv.ulisboa.pt (J.M.); catia.marques@ulusofona.pt (C.M.); 2Associate Laboratory for Animal and Veterinary Sciences (AL4AnimalS), 1300-477 Lisbon, Portugal; 3Faculty of Veterinary Medicine, Lusófona University, 1749-024 Lisbon, Portugal; 4Molecular Veterinary Diagnostic Laboratory-Genevet, 2790-140 Carnaxide, Portugal

**Keywords:** companion animals, Enterobacterales, carbapenemase detection methods

## Abstract

The dissemination of antimicrobial-resistance is a major global threat affecting both human and animal health. Carbapenems are human use β-lactams of last resort; thus. the dissemination of carbapenemase-producing (CP) bacteria creates severe limitations for the treatment of multidrug-resistant bacteria in hospitalized patients. Even though carbapenems are not routinely used in veterinary medicine, reports of infection or colonization by carbapenemase-producing Enterobacterales in companion animals are being reported. NDM-5 and OXA-48-like carbapenemases are among the most frequently reported in companion animals. Like in humans, *Escherichia coli* and *Klebsiella pneumoniae* are the most represented CP Enterobacterales found in companion animals, alongside with *Acinetobacter baumannii*. Considering that the detection of carbapenemase-producing Enterobacterales presents several difficulties, misdiagnosis of CP bacteria in companion animals may lead to important animal and public-health consequences. It is of the upmost importance to ensure an adequate monitoring and detection of CP bacteria in veterinary microbiology in order to safeguard animal health and minimise its dissemination to humans and the environment. This review encompasses an overview of the carbapenemase detection methods currently available, aiming to guide veterinary microbiologists on the best practices to improve its detection for clinical or research purposes.

## 1. Introduction

Carbapenems are β-lactam antibiotics with broad antimicrobial spectrum. With the emergence of Extended Spectrum β-lactamases (ESBLs), carbapenems became the antibiotics of last resort for treatment of human patients with ESBL-producing Enterobacterales infections [1]. Although carbapenems are not hydrolysed by most β-lactamases, their effectiveness was seriously compromised by the emergency of carbapenem-hydrolysing enzymes, the carbapenemases [1,2]. The most important carbapenemases belong to three different Amber classes [2]: (i) class A, including the KPC, IMI/NMC, SFC, GES type enzymes [1,2]; (ii) class B, including VIM, IMP, and NDM metallo-β-lactamases (MBL) [3]; and (iii) class D, including OXA-48-like type enzymes [4].

Regulation on the use of carbapenems in animals varies worldwide and they do not belong to the OIE List of Antimicrobial Agents of Veterinary Importance [5]. According to the European Medicine Agency categorization of antibiotics for animal use, carbapenems are included in category A (“Avoid”), meaning they are not authorized for use in veterinary medicine in the European Union (EU), except in exceptional clinical cases in companion animals, under the cascade according to Article 112 of the veterinary medicinal products Regulation 2019 of the European Union Legislation [6]. Reports of carbapenemase-producing (CP) and carbapenem-resistant Enterobacterales (CRE) detection among companion animals are emerging worldwide (Table 1). The identification of CP bacteria in companion animals, which have significant direct contact with humans, has raised public health concern as animals may constitute an important reservoir of carbapenems resistance genes and contribute to its dissemination [7]. Very recently, the building of an European Antimicrobial Resistance Surveillance network in veterinary medicine (EARS-Vet) has been reported [8]. However, carbapenem resistance epidemiology remains quite unknow, as, unlike in human medicine, no global surveillance protocol is currently in place for companion animal veterinary medicine. Furthermore, the detection of CP bacteria relying on antimicrobial susceptibility testing alone (AST) presents several pitfalls leading to its possible miss detection in veterinary medicine. Appropriate monitoring and detection of antimicrobial resistance against these critically important antimicrobials in veterinary medicine is of the utmost importance to avoid treatment failure and prevent its dissemination to humans and the environment. However, there is a lack of recommendations directed specifically to the veterinary medicine needs in the published literature.

In this review, an updated overview of the current methods available for the detection of CP bacteria directed at veterinary medicine will be made aiming to guide veterinary microbiologists on the best practices to improve carbapenemase detection for clinical AST reports or even research purposes.

## 2. Carbapenemase-Producing Bacteria in Companion Animals

To our best knowledge, more than 25 reports of CP bacteria in dogs and cats have been published worldwide. These include, both infection and colonization CP isolates harbouring KPC, VIM, IMP, NDM, or OXA β-lactamases (Table 1).

Briefly, three studies detected KPC-producing *Escherichia coli* and *Klebsiella pneumoniae* from dogs in Brazil and in *Enterobacter xiangfangensis* from a dog in the United States [9,10,11]. A IMP-4-enzyme in *Salmonella* isolates was recovered from a cat’s faecal samples in Australia [12], VIM-2 in *Pseudomonas aeruginosa* from dogs with pyoderma and otitis in South Korea [13] and VIM-1 in *K. pneumoniae* from dogs in Spain [14]. A number of NDM-5-producing *E. coli* have been found in dogs and cats [15,16,17,18,19,20,21,22], one NDM-1-producing *Acinetobacter radioresistens* was detected in a dog, six NDM-1-producing *E. coli* from dogs and cats in the United States, two NDM-1-producing *E. coli* from a dog in China, and finally one NDM-9 from a farm dog in China [23,24,25]. Several OXA-48-like carbapenemase-producing *E. coli*, *K. pneumoniae, Klebsiella oxytoca,* and *Enterobacter cloacae* isolates were recovered from dogs, cats, and horses, representing one of the most frequent carbapenemases detected in companion animals alongside with NDM-5 (Table 1) [17,26,27,28,29,30,31,32]. In addition, OXA-23- and OXA-66-producing *Acinetobacter baumannii* were isolated from clinical samples from dogs and cats [23,33,34].

Interestingly, although the detection of CP bacteria in companion animals dates to at least 2009, detection methods vary widely between studies, with the use of selective culture media being the most frequent for the detection of commensal CP isolates, while antimicrobial susceptibility testing alone (AST) is the main method used for the detection of CP isolates in infection cases (Table 1). Another important finding is that most CP bacterial species isolated from companion animals belong to the priority 1 (“critical”) category within the WHO priority pathogens list [35], thus highlighting the importance of properly monitoring and effectively detecting these carbapenem resistance mechanisms in companion animals.

**Table 1 antibiotics-11-00533-t001:** Carbapenemases found in companion animals across the world.

Enzyme	Year	Country	Host	Source	Bacterial Species	Detection Methods	Refs.
IMP-4	2016	Australia	Cats	Commensal	*Salmonella* *enterica* serovar Typhimurium	AST	[12]
KPC-2	2018	Brazil	Dog	Infection (UTI)	*Escherichia coli*	Imipenem synergy test, modified Hodge testing, PCR	[9]
KPC-2	2021	Brazil	Dog	Infection (UTI)	*Klebsiella* *pneumoniae*	Imipenem synergy test, AST	[10]
KPC-4	2018	USA	Dog	Infection (UTI, SSTI)	*Enterobacter xiangfangensis*	Biochemical Tests	[11]
NDM-1	2013	United States	Dogs, Cats	Infection (SSTI, UTI)	*Escherichia coli*	AST	[24]
NDM-1	2017	China	Dogs	Commensal	*Escherichia coli*	Selective culture media	[16,25]
NDM−1	2018	Italy	Dog	Commensal	*Acinetobacter* *radioresistens*	Selective culture media	[23]
NDM-5	2016	Algeria	Dogs	Commensal	*Escherichia coli*	PCR	[17]
NDM-5	2017	China	Dogs	Commensal	*Escherichia coli*	Selective culture media	[16]
NDM-5	2019	United Kingdom	Dog	Infection (SSTI)	*Escherichia coli*	AST	[19]
NDM-5	2018	Finland	Dogs	Infection (Otitis externa)	*Escherichia coli*	AST followed by modified Hodge testing, UV spectrometric detection of imipenem hydrolysis	[18]
NDM-5	2021	Italy	Dog	Infection (UTI)	*Escherichia coli*	Meropenem synergy test	[15]
NDM-5	2018	United States	Dog	Infection (URTI)	*Escherichia coli*	AST	[20]
NDM-5	2018	United States	Dogs, Cats	Infection (UTI, URTI)	*Escherichia coli*	AST	[22]
NDM-5	2018	South Korea	Dog, Cat	Commensal	*Escherichia coli*	AST, PCR	[21]
NDM-9	2017	China	Dog	Commensal	*Escherichia coli*	Selective culture media	[16]
OXA-48	2009–2010	Germany	Dogs, Cats, Horses	Infection	*Escherichia coli*, *Klebsiella* *pneumoniae*, *Enterobacter* *cloacae*	Selective culture media for cephalosporin resistance, PCR	[36]
OXA-48	2013	Germany	Dog	Commensal, Infection (UTI, SSTI, URTI, CRBSI)	*Klebsiella* *pneumoniae*, *Escherichia coli*	AST	[29]
OXA-48	2016	United States	Dogs, Cats	Infection (UTI, SSTI, Genital tract)	*Escherichia coli*	AST	[31]
OXA-48	2016	Algeria	Dogs	Commensal	*Escherichia coli*	PCR	[17]
OXA-48	2017	Algeria	Dogs, Cat, Horses, Pet birds	Commensal	*Enterobacter* *cloacae*, *Escherichia coli*, *Klebsiella* *pneumoniae*	Selective culture media	[32]
OXA-48	2017	France	Dog	Commensal	*Escherichia coli*	Selective culture media	[30]
OXA-48	2018	Germany	Dogs, Cats, Horses	Infection (UTI, SSTI, genital tract, otitis, URTI)	*Klebsiella* *pneumoniae*, *Enterobacter* *cloacae*, *Escherichia coli*, *Klebsiella oxytoca*	Selective culture media	[28]
OXA-181	2018	Switzerland	Dogs, Cats	Commensal	*Escherichia coli*	Selective culture media	[26]
OXA-181	2020	Portugal	Dog	Commensal	*Escherichia coli*	Selective culture media and AST	[27]
OXA-181	2021	Portugal	Cat	Infection (SSTI)	*Klebsiella* *pneumoniae*	Selective culture media and AST	[37]
OXA-23	2014	Portugal	Cat	Infection (UTI)	*Acinetobacter* *baumannii*	AST	[33]
OXA-23	2017	Germany	Dogs, Cats	Infection (UTI, suppurate inflammation)	*Acinetobacter* *baumannii*	Selective culture media	[34]
OXA−23	2018	Italy	Dogs, Cats	Commensal	*Acinetobacter* *baumanni*	Selective culture media	[23]
OXA-66	2017	Germany	Dogs, Cats	Infection (UTI, SSTI, URTI, CRBSI, suppurate inflammation)	*Acinetobacter* *baumannii*	Selective culture media	[34]
VIM-1	2016	Spain	Dog	Commensal	*Klebsiella* *pneumoniae*	Selective culture media, Meropenem synergy test	[14]
VIM-2	2018	South Korea	Dog	Infection (SSTI)	*Pseudomonas* *aeruginosa*	AST	[13]

AST, antimicrobial susceptibility testing; CRBSI, catheter-related bloodstream infection; SSTI, skin soft tissue infection; URTI, upper respiratory tract infections; UTI, urinary tract infection.

## 3. Phenotypic Characteristics of Carbapenemases and Their Genetic Background in Isolates from Companion Animals

The β-lactam resistance phenotype of CP isolates can vary depending on the type of carbapenemase and its hydrolysing activity (Table 2).

### 3.1. Serine Carbapenemases

Serine carbapenemases of molecular (Ambler) class A corresponds to the KPC, IMI/NMC, SFC, and GES enzymes that have a hydrolytic mechanism involving an active site serine at position 70 (Ambler numbering of class A β-lactamases), conferring resistance to first-, second-, and third-generation cephalosporins, imipenem, and meropenem [2].

Class A carbapenemases have been rarely detected in companion animals, the KPC enzyme being the only one reported until now from dogs with UTI and SSTI (Table 1). In *K. pneumoniae* and *E. coli* from dogs, the *bla*_KPC-2_ gene was found in Tn*4401* transposons contained in IncN plasmids [9,10] and the *bla*_KPC-4_ gene was detected in an IncHI2 plasmid in the context of Tn*4401b* transposon in *E. xiangfangensis* isolated from a dog’s clinical samples [11].

### 3.2. Metallo-β-Lactamases

Class B carbapenemases have a critical clinical significance due to their ability to hydrolyse all β-lactams (Table 2) [3,40]. So far, more than 50 allelic β-lactamase-conferring imipenem resistance (IMP) variants are listed at GenBank DNA sequence database. However, only IMP-4 has been reported among companion animals, namely, cats, in *Salmonella enterica* serovar Typhimurium (Table 1) [12]. The IMP-4 coding gene was located on a gene cassette (*bla*_IMP-4_-*qacG-aacA4-catB3*) in a class 1 integron, associated with a conjugative plasmid IncHI2, also carrying other resistance genes, such as *tetA* (mediating resistance to tetracycline), *aac* (resistance to aminoglycosides), *cat* (chloramphenicol resistance), *sul* (sulphonamide resistance), *bla*_OXA_ (different serine oxacillinases), and *bla*_TEM-1_ (narrow-spectrum β-lactamases) [12].

Verona Integron-encoded Metallo-β-Lactamase (VIM) enzymes are the second most common Class B carbapenemase detected in companion animals (Table 1). VIM-1 and VIM-2 were described in *K. pneumoniae* and *P. aeruginosa* isolates from dogs, respectively; both located in class 1 integrons incorporated on untyped plasmids [13,14].

The *bla*_NDM_ genes pose a serious public health concern, since most common plasmids associated with its spread often have various antibiotic resistance genes resulting in multidrug resistance phenotypes [18,19,41]. Until now, 28 variants have been described, with resistance against all β-lactams expect monobactams [42]. In companion animals only NDM-1 and NDM-5 have been described so far (Table 1), the latter being more frequent. For one metallo-β-lactamase NDM-1, the encoding gene was located in a transposon Tn*125* (composed of *bla*_NDM-1_-*ble_MBL_*-*trpF*-*TAT*-*cutA1*-*groES*-*groEL*-*insE*-*Δpac* genes between a pair of IS*Aba125*), integrated in the chromosome of an *A. radioresistens* isolated from a dog in Italy [23]. This Tn*125* transposon usually encompasses *bla*_NDM_ genes with two flanking IS*Aba125* elements, and in companion animals it was also found in *bla*_NDM-5_ carrying strains [16,23]. A NDM-1-producing *E. coli* isolate harboured *bla*_NDM-1_ in another genetic region, which was not flanked by IS*Aba125* elements downstream of the resistance gene [25]. NDM-5 metallo-β-lactamase differs from NDM-1 by four amino acids and has been found in the chromosome of an integrated IncF plasmid, from an *E .coli* isolate causing skin and soft tissue infection on a dog in the United Kingdom [19]. In the United States, the *bla*_NDM-5_-encoding gene has been found on IncFII-type plasmids [20,22], whereas in South Korea it was described in an IncX3-type plasmid [21] with the surrounding genetic environment of IS*Aba125-bla*_NDM-5_-*ble_MBL_*-*trpF*-*TAT*-IS*CR26*. 

### 3.3. Oxacillinases

The class D, carbapenem-hydrolysing OXA-48 and its variants, namely, OXA-181, are one of the most common in veterinary settings (Table 1). The OXA-181 variant weakly hydrolyses both carbapenem and extended-spectrum cephalosporins and differs from OXA-48 at four amino acid substitution, yet its kinetic properties appear broadly similar to OXA-48 [43,44,45]. These enzymes can be associated more with different β-lactam hydrolysis profiles than the other serine-metallo-β-lactamases, making its accurate detection difficult. By possibly being susceptible in vitro to meropenem and imipenem (Table 2), two widely used surrogates to identify carbapenem resistance in clinical microbiology, carbapenem-resistant bacteria harbouring OXA-48-like carbapenemases may easily be misdiagnosed as ESBL-producers, which may lead to treatment failure. OXA-48-coding genes in CP isolates have been associated with no other resistance genes; or with extended-spectrum β-lactamases coding genes, thus conferring either low or high MIC against carbapenems [46]. High-level resistance to carbapenems has also been observed [31] that may be associated with the combination of these carbapenemases with outer membrane lack of permeability [47]. Importantly, regardless of the carbapenem susceptibility profile detected in vitro, carbapenem therapy is not reliable against OXA-48-like-producing bacteria [45].

In companion animals, the *bla*_OXA-48_ gene has been commonly observed on pOXA-48a plasmid, a self-conjugative IncL/M plasmid [28,29,30]. This plasmid has a high conjugation rate, therefore, it can be transferred at a very high frequency across Gram-negative bacteria [41,48]. Flanking the *bla*_OXA-48_ gene is the Tn*1999* composite transposon, which cooperates in mobilizing pOXA-48a or closely related plasmids [44,48].

The *bla*_OXA-181_ gene was found to be part of the transposon Tn*2013,* inserted at the downstream region of IS*Ecp1,* which is a very efficient genetic vehicle for spreading ESBL genes, namely, the *bla*_CTX-M-15_ gene [49]. The *bla*_OXA-181_ gene has been frequently identified in IncX3 plasmids [26,27].

The frequency of OXA-48-like-producing bacteria in companion animals (Table 1) and its frequent association with mobile genetic determinants that facilitate its dissemination, highlight the importance of monitoring this resistance mechanism in companion animals. Furthermore, the possible misdiagnosis of OXA-48-like-producing bacteria when using meropenem and imipenem as surrogates may lead to underestimating its frequency and the epidemiological role of companion animals as reservoirs.

The *bla*_OXA-23_ gene has been reported coming from *A. baumannii* isolates (Table 1). This gene is often located on transposon Tn*2006*, but has also been identified in transposon Tn*2008* in animals isolates [23,34]. The *bla*_OXA-23_ is usually flanked between IS*Aba1* insertion sequences, known to promote the expression of *bla*_OXA-23_ and *bla*_OXA-51-like_ genes in *A*. *baumannii* for an elevated level sufficient to display carbapenem resistance [23,50]. In addition to carbapenems, the OXA-23 enzymes can hydrolyse cephalosporins, aminopenicillins, piperacillin, oxacillin, and aztreonam (Table 2) [4].

## 4. Methods for Detection and Identification of Carbapenemases

Detection of CP bacteria has proven to be a difficult task, as it cannot solely be based on the resistance profile observed during AST [51]. Usually, an elevated MIC against a carbapenem is a marker for testing for carbapenemase production. However, some CP isolates have low carbapenem MICs, being susceptible according to EUCAST and CLSI guidelines, such as OXA-type carbapenem-hydrolysing class D β-lactamases [26,27,29,30].

For such reason, it is important for veterinary diagnostic laboratories to employ specific tests to correctly identify CP bacteria during routine microbiology procedures. The accurate detection of carbapenem resistance is key to improve animal health; and to minimize its dissemination to humans and the environment. A variety of methodologies and tests are available for this purpose, which vary in its practicality and in the technical expertise required.

### 4.1. Selective Culture Media

Several different selective culture media are available for the detection of CP isolates. The most common ones are: SUPERCARBA Medium (CHROMagar™, Paris, France); CRE Agar (Brilliance™ Oxoid, Thermofisher Scientific Illkirch, Dardilly, France); ChromID CARBA Smart (Biomerieux, Marcy l’Etoile, France) and CHROMagar™ KPC/OXA-48 (CHROMagar™) (Table 3) [52,53,54]. All these culture media have chromogenic molecules in their composition, allowing for a rapid presumptive species identification after overnight incubation. Several studies have been conducted, comparing and evaluating the performance of these selective culture media. The SUPERCARBA medium seems to have the higher sensitivity of all, ranging from 95.6% to 96.5%, with 100% sensitivity for KPC and OXA-48 producers [55,56]. However, its specificity decreases to 60.7%, as it also detects non-carbapenemase isolates that are carbapenem-resistant due to ESBL/AmpC overexpression in combination with porin loss [55]. Regarding the CRE Agar medium, it has sensitivity of 78% and a specificity that variates from 60 to 66% [52]. A study conducted in Germany has shown that ChromID CARBA Smart fails to detect CP isolates with low carbapenem resistance, when comparing the same isolates plated on MacConkey agar supplemented with 1 mg/L of cefotaxime and 0.125 mg/L of meropenem [57]. This culture medium presents a sensitivity of 91% and a specificity that variates from 76 to 89% [52]. The CHROMagar™ KPC medium presents a sensitivity of 100% [53], with a positive predictive value (PPV) of 100% for KPC producers and negative predictive value (NPV) of 98.8%, whereas in the same comparison study, MacConkey agar supplemented with 1 mg/L of imipenem yielded 94.7% PPV and 88.6% NPV, having failed to detect 10 positive isolates [58]. Another specific carbapenemase selective medium is CHROMagar™ OXA-48; however, its sensitivity is suboptimal (75.8%) in direct sampling, only increasing to 90.9% when performed after an enrichment method. Nonetheless, its specificity is 99.3% [54].

Apart from CHROMagar™ KPC/OXA-48, none of the other media can accurately identify the specific *bla* gene responsible for causing resistance against carbapenems. Regardless of this, all isolates grown in these selective media must be confirmed as CP with subsequent molecular testing [59].

### 4.2. Biochemical Tests

Biochemical tests are relatively quick and easy to use (Table 3). To the best of our knowledge, the Rapidec^®^ CarbaNP (Biomérieux, Marcy l’Etoile, France) was the first commercial kit of its kind offering a positive result under two hours. Positive results occur due to colour shifting from pH alteration as consequence of carbapenem hydrolysis [60]. It has 100% sensibility and specificity for Enterobacterales [60,61]. A study conducted by Tijet et al. reported a decreased sensibility (72.5%) on account of mucoid isolates and/or isolates harbouring low carbapenemase activity genes, such as OXA-48 and GES-5 [62]. When using the commercial version of the CarbaNP test, some difficulties in results interpretation due to colour shifting have been reported [63].

As a cheaper alternative to the CarbaNP test, the carbapenem inactivation method (CIM) is available. CIM is also based on carbapenem hydrolysis. In this test, a disc of meropenem 10µg was immersed in a bacterial suspension of the isolate to be tested and incubated for a minimum of two hours at 35 °C. A Mueller–Hinton Agar plate is inoculated with a known susceptible *E. coli* strain prior to disc placement [64]. The turnaround time is approximately eight hours, and if the bacterial isolate produced carbapenemase, the meropenem in the susceptibility disk was inactivated allowing uninhibited growth of the susceptible indicator strain. Disks incubated in suspensions that do not contain carbapenemases yielded a clear inhibition zone.

The Blue Carba test (BTC) is another test that also gives results within 2 h. Similar to CarbaNP, it is based on imipenem hydrolysis by CP bacteria, which leads to colour changes due to pH alteration in case of a positive result [65]. It has a sensibility and specificity of 100%, with additional advantages: use of colonies grown on Mueller–Hinton Agar; it is cheaper as it does not use imipenem monohydrate but Tienam^®^ (imipenem/cilastatin, Merck Sharp & Dohme, Campinas, Brazil); and it was validated against OXA-type carbapenemases [65]. However, in a study conducted by Pasteran et al., the OXA-type carbapenemase detection using this method had a sensibility and specificity of 97% and 96%, respectively. On the other hand, isolates with low imipenem MICs were correctly identified [66].

The β CARBA Test™ (Bio-Rad, Marne la Coquette, France) also relies on colour changing for result interpretation. It has a sensibility of 84.9% and specificity of 95.6%, having failed to detect non-KPC Ambler class A carbapenemases [67].

Besides allowing a short turnaround time, another advantage of these biochemical tests is the fact that, unlike PCR, these are not targeted to any specific carbapenemase groups. Therefore, they allow the detection of carbapenemase activity of yet undiscovered resistance genes.

### 4.3. Disc Diffusion Methods

This method requires the use of meropenem discs and meropenem discs supplemented with different inhibitors to detect and identify carbapenemases (Figure 1) [63].

Meropenem synergy with phenyl boronic acid is indicative of the presence of Ambler class A KPC. Meropenem synergy with EDTA plus dipicolinic acid is indicative of the presence of an MBL. Detection of a positive synergy with a disc of cloxacillin plus phenyl boronic acid indicates carbapenem resistance due to porin loss or AmpC overexpression [68]. A zone diameter increase of ≥4 mm around the discs containing inhibitors, in comparison with the disc with meropenem alone, is considered to be a positive synergy result for phenyl boronic acid, whereas an increase of ≥5 mm is considered to be a positive synergy result for EDTA/dipicolinic acid and cloxacillin/phenyl boronic acid (Figure 1) [63,68,69].

To detect OXA-48-like CP bacteria, it is recommended to add a temocillin disc (30 µg) to the group of synergy tests (Figure 1), due to its weak meropenem hydrolysis [4,38,63,69]. Temocillin lacks activity against Gram-positive bacteria as well as non-fermenters [70]. Temocillin high-level resistance phenotype is proposed as a marker for OXA-48-like producers. However, this marker is not specific for OXA-48-like carbapenemases, as other resistance mechanisms may confer this phenotype; therefore, the presence of carbapenemases must be confirmed using complementary tests in all isolates showing a zone diameter ≤10 mm [63,69]. In a study conducted by van Dijk et al., this detection method had a 100% sensibility and specificity, with all positive isolates having a zone diameter ≤10 mm [69]. Nevertheless, the authors alert to its incapability of detecting MBL in combination with OXA-48 producers.

The main disadvantage in using disc diffusion methods for carbapenemase screening is the turnaround time of approximately 18 h of incubation, whereas biochemical tests have a turnaround time of 2 h. However, they are low cost compared with biochemical testing.

The Hodge modified test used to be an option as a phenotypic method, but due to its dubious results and low sensibility/specificity, its use has since been advised against by EUCAST and CLSI [63,71].

### 4.4. Lateral Flow Assays

Some immunochromatographic assays are available to readily identify suspecting colonies grown in non-specific media. Currently, there are numerous options for lateral flow assays, yet the most commonly used seem to be the OXA-48 K-set, KPC K-set, Resist-3 O.K.N K-set, and Resist-4 O.K.N.V. (CorisBio Concept, Gembloux, Belgium). Resist-3 O.K.N K-set is a multiplex assay, detecting OXA-48, KPC, and NDM-like enzymes [72]. The sensitivity and specificity is 100% for all cassettes [72,73,74]. Although these tests were designed to be used with bacterial inoculum, studies have evaluated the multiplex efficacy in positive blood culture bottles (BacT/ALERT, Biomérieux). The multiplex system is compatible with blood culture bottles, albeit it showed weak signal bands for NDM-like enzymes positive isolates; and the positive signal was also influenced by the blood volume used [75]. The fourth version of the test, the Resist-4 O.K.N.V., added the detection of VIM-like enzymes to the previous Resist-3 O.K.N K-set. The test maintains its 100% sensitivity to OXA-48-like and KPC enzymes, as well as VIM, but it decreases to 83.3% regarding NDM producers [76]. Previous reports have shown 100% sensitivity for NDM producers detected using Resist-4 O.K.N.V.; however, this could be as in some of those studies only Enterobacterales were evaluated [73,77]. Another possible explanation is the impact that the morphological characteristics of the colony used in the test can have in its accuracy. The NG-Test^®^ CARBA 5 (Hardy Diagnostics, Santa Maria, CA, USA) is another available commercial kit, which can additionally detect IMP enzymes. Similar to the previously described tests, NG-Test^®^ CARBA 5 has 100% sensitivity and sensibility [78]. Recently, the Resist-5 O.K.N.V.I. cassette was also launched, but compared to its homologous, it has sensitivity of 98.4% and specificity of 100% [79]; however, not many comparative studies have been conducted and none have compared both rapid tests. Overall, these immunochromatographic assays are useful to be used as a screening method in routine microbiology when the isolated bacteria are suspected to be a carbapenemase producer. Nonetheless, positive results should always be confirmed with PCR targeting for the most common carbapenemase genes [59].

### 4.5. Molecular Testing

Molecular techniques are mostly based on PCR, and may be followed by sequencing if needed for precise identification of a specific carbapenemase, rather than just its group (e.g., KPC-type, IMP-type, VIM-type, NDM-type, and OXA-type) [59].

Nowadays, PCR assays are becoming a routine method in many veterinary clinical diagnostic laboratories. This molecular testing remains the reference standard for the identification and differentiation of carbapenemases, recommended by guidelines and expert groups [63,80].

PCR assays performed on genomic DNA for the detection of carbapenemase genes are easily available in the literature, including multiplex PCRs, and can give results within 4–6 h [59,81].

Nevertheless, these PCRs require the acquisition and manipulation of CP control strains to be used as DNA positive controls. Nowadays, there are real time-PCR fully automated systems that allow the detection of *bla*_VIM_, *bla*_NDM_, *bla*_IMP_, *bla*_OXA-48_, *bla*_KPC_, *bla*_OXA-23_, *bla*_OXA-58_, *bla*_OXA-24_, and ISAba1-associated *bla*_OXA-51_ carbapenemase genes and the colistin-resistant *mcr-1* gene, such as the Novodiag^®^ CarbaR+ (Mobidiag, Espoo, Finland). The Novodiag^®^ CarbaR+ test can be applied to fresh bacterium isolates or directly from rectal swabs, having a sensitivity and specificity of 98.2% and 99.7%, respectively [82]. Results are available after 1 h approximately, with only 5 min hands-on preparation of samples, which might be crucial when dealing with critically ill patients. When rectal swabs are directly analysed, the sensitivity decreases only slightly to 97.8% and specificity to 98.6%, revealing its usefulness to rapidly detect colonized patients. Compared to other already available tests within the same category, the main advantage of Novodiag^®^ CarbaR+ is that the panel tested is comprised of resistance genes associated with carbapenem resistance in *Pseudomonas* spp. and *Acinetobacter* spp. An alternative to this automated system is the Xpert^®^ Carba-R (Cepheid, Sunnyvale, CA, US), which is quite similar in terms of functioning but only tests for the main 5 carbapenemase groups—*bla*_VIM_, *bla*_NDM_, *bla*_IMP_, *bla*_OXA-48_, and *bla*_KPC_. PCR and real-time PCR are standard techniques, widely used by the scientific community.

Similarly commercial PCR kits are also available, such as the Check-MDR CT103XL (Check-Points Health, Wageningen, The Netherlands) DNA microarray assay, capable of detecting a wide range of carbapenemase (KPC, GES, IMP, VIM, NDM, OXA-23-like; OXA-24-like, OXA-48-like, and OXA-58-like) with an accuracy of 94.2% [83]. The principle of the Check-Points diagnostic system is based on DNA amplification followed by amplicon detection in a tube microarray [83].

Another type of molecular commercial kit is the eazyplex^®^ SuperBug CRE (AmplexDiagnostics, Gars-Bahnhof, Germany), a loop-mediated isothermal amplification (LAMP) method that can be used for direct screening of KPC, VIM, NDM, and OXA-48-like carbapenemases on rectal swab and urine samples in 20 min as well as confirmation from positive blood culture and culture plate in 15 min. This LAMP assay has shown a sensitivity from 95.2% to 100% with a specificity of 97.9% [84].

An alternative promising molecular commercial methodology is the hybridization technology by Luminex xMAP (Multi-Analyte Profiling, Austin, TX, USA), which although it does not have a specific panel for carbapenemase detection available, one can create their own personalised panel. In a study by Bilozor et al., this system had a sensitivity >95% in detecting KPC, IMP, VIM, NDM, and OXA-48-like carbapenemases when using a tailored panel [85].

Nonetheless, specific equipment and experienced staff are required for theses molecular-based technologies, which might be seen as a disadvantage to smaller microbiology laboratories. Furthermore, it should be noted that only the carbapenemases targeted by each specific assay will be detected.

Whole genome sequencing (WGS) is a state-of-the-art methodology with promising applications for medical microbiology [86]. WGS allows fast and accurate identification and typing of pathogens with the highest possible discriminatory power currently available for effective surveillance and outbreak detection. In addition, WGS of bacterial genomes provides information regarding antimicrobial resistance determinants; virulence and pathogenicity determinants; in addition to providing data for the discovery of new genetic determinants [86]. Thus, efforts are being made by the scientific community to use WGS in the routine laboratory workflow to improve the diagnostic turnover and retrieve information that might replace older routine procedures that are time consuming and expensive. Furthermore, the use of untargeted metagenomic next-generation sequencing (mNGS) from clinical samples is also considered very promising. Although still being improved, mNGS may revolutionize the diagnosis of infectious disease in the future, since, once optimized, it may allow the simultaneous identification of viruses, bacteria, fungi, and parasites in a single assay [87]. The main disadvantage in using these techniques, nowadays, lies in the need of a multidisciplinary team of personnel specialized in WGS/NGS and bioinformatics [86]. Data analyses should be performed by staff members who have been trained to use commercial or open-source software tools to extract the appropriate information from the large amount of sequence data that is generated, and ultimately deliver clinically relevant information to the clinicians. To become truly accessible as a future everyday routine diagnostic tool, new user-friendly software platforms need to be developed so that information may be retrieved easily without the need of extensive technical and bioinformatic skills.

### 4.6. Mass Spectrometry Analysis

Matrix-assisted laser desorption ionization-time of flight mass spectrometry (MALDI-TOF MS) is an analysis method where the material is ionized within a high vacuum chamber, and accelerated in an electric field. Being widely used for species identification, it can also be used to detect carbapenemase production through enzyme detection. Prior to testing, it is necessary to establish the mass spectrum of a pure carbapenem [88]. Some carbapenemases have fast carbapenem hydrolysis activity compared to others, which are slower, making it necessary to have several runs and specs to achieve a reliable result. Several protocols for detection have been described, with the lack of standardization being a problem when trying to implement this method [89]. MALDI-TOF MS requires an even higher level of staff expertise for result interpretation when compared to most of the previously described methods. It offers a fast response compared to molecular and disc diffusion methods, but it is expensive to buy the necessary equipment if not already in use and it does not detect other carbapenem resistance mechanisms such as porin loss.

## 5. Transmission Potential

The regular and close contact between companion animals and humans provides excellent opportunities for interspecies transmission of resistant bacteria and their resistance genes in either direction [18,90,91,92]. Hence, the increasing trends and prevalence of carbapenem-resistant bacteria observed in many companion animals is of major public health concern as companion animals could be reservoirs of CP bacteria, thus acting as direct players in the transmission of these resistant bacteria to humans [7].

The European Medicine Agency and its Antimicrobial Working Party have already warned about the indirect hazard associated with carbapenem-resistant bacteria from companion animals to public health in its reflection paper [7]. Since then, sharing of clinical NDM-5-producing multidrug-resistant ST167 *E. coli* between dogs and co-habiting human was reported in a Finland study [18]. Moreover, in a Chinese study across farming sectors, common NDM-positive *E. coli* strains were identified among farms, flies, dogs, and farmers [16], providing additional scientific support regarding concerns not only about the transfer of resistance between companion animals and humans, but also about their potential role as reservoirs for environmental contamination [16,18,90]. Furthermore, the similarity of carbapenem-resistant clonal lineages isolated from companion animals and humans worldwide, and its genetic features, suggests an interspecies exchange of resistant-bacteria or resistance genes located at mobile genetic elements [26,27,28,29].

There is a big concern regarding carbapenemases following the same exponential spread as ESBL-producing bacteria, where reports of transmission between companion animals and humans are numerous worldwide [91,92,93,94]. ESBL-producing Enterobacterales can serve as a model for the spread of CP bacteria because the same bacterial species are involved, and the resistance genes are also carried on plasmids [95].

Studies have shown that bacteria causing infection in dogs and cats, were increasingly resistant to the antimicrobials most widely used for animal treatment [94,96,97]. Although the use of carbapenems is not currently licensed for companion animals, it has been reported and it is regulated in the EU under the cascade prescribing [6,98]. Nevertheless, the dissemination of CP bacteria to companion animals is likely one of an anthropogenic nature, since carbapenems are essentially used in human medicine. Once colonizing companion animals, one must keep in mind that the exposure to systematic broad-spectrum antimicrobials approved for veterinary use, including β-lactams, are likely to co-select and facilitate the propagation of CP bacteria within the companion-animal population, thus further highlighting the relevance of monitoring these resistance mechanisms in veterinary microbiology.

To foster antimicrobial stewardship and, consequently, the reduction in the emergence of resistance, a prudent use of critically important antimicrobials for human medicine, such as fluoroquinolones, aminoglycosides, and third-generation cephalosporins, is needed. Furthermore, the risk of increasing selection pressure for the maintenance of CP bacteria in companion animals gut and their potential transfer to humans, needs a severe restriction or elimination of carbapenems use in veterinary medicine worldwide [98,99]. Not only are pet owners at risk of acquiring these resistance strains by interspecies transmission due to direct contact and/or indirectly via the common environment, but, also, veterinary personnel, veterinary students, or trainees are a professional hazard group. As important CP bacteria also present animal health risks due to treatment failure. Therefore, it is vital to implement systematic monitoring programs in veterinary laboratories to screen for carbapenem resistance in a One Health perspective. The screening of CR and CP bacteria should be conducted in all companion animal samples submitted to culture and AST regardless of clinical presentation and animal species. The most reliable detection methods should be preferred according to each laboratory technical and financial availability. Figure 2 summarizes a possible workflow that can be adapted to veterinary diagnostic laboratories.

A pressing action is required to reduce the public and animal health hazard posed by the emergence of carbapenem-resistant bacteria isolated from companion animals. In summary, these safety measures should be taken in consideration [98,100]:Achieving the principles of prudent use of antibiotics in veterinary practice to ensure that carbapenems are used only in the very few cases that lack other suitable alternatives based on culture and AST;Include the systematic screening for carbapenem resistance in veterinary microbiology laboratories;Surveillance and monitoring for the presence of genes encoding resistance to critically important antimicrobials, such as carbapenems;Appropriate hygiene practices after handling animals both in domestic and health care settings;Infection control measures when dealing with companion animals with infections caused by carbapenem-resistant strains that include isolation of infected animals.

Communication between all the specialists involved in human and veterinary medicine should be established in a One Health approach to develop a universal strategy that scientific and non-scientific audiences can follow.

## 6. Conclusions and Final Remarks for Veterinary Medicine

In veterinary medicine, screening for carbapenemase-producing bacteria is not usually performed, and frequently relies on the use of meropenem and imipenem as surrogates in AST. However, in past years, many reports on the presence of genes encoding for carbapenemases in companion animals have been made, as well as its direct transmission to humans. Accurate detection of CP bacteria is essential for infection control purposes and particularly to minimize the spread of its resistant determinants that are known to cause a major health impact by limiting antimicrobial therapy.

Even though all the detection methods presented here are applied and have been evaluated in Human Medicine, not all are useful in Veterinary Medicine. For example, investing in automated PCR machines or a Mass Spectrometer is not yet viable for most laboratories, whether due to the expected low volume of positive samples, the need for experienced staff, or the required financial investment in expensive equipment and consumables.

Although molecular identification of carbapenemase encoding genes is the gold standard, the phenotypic detection of carbapenem resistance is a feasible alternative for routine diagnosis. Screening of CP bacteria on clinical veterinary laboratories can be made quite easily and affordably, by implementing commercially available CP selective culture media in the veterinary microbiology routine workflow. Ideally, all Enterobacterales isolates should be plated despite AST results. Furthermore, the high frequency of OXA-48-like CP bacteria reported in companion animals and the pitfalls in its detection should prompt the inclusion of temocillin in the routine AST of samples from companion animals or the use of selective culture media with high sensibility and specificity for OXA-48-like carbapenemases. Other methods, such as the biochemical test or immunochromatographic lateral flow assays, may be useful in laboratories with a high case load of suspected CR infections, which is currently not yet the case for veterinary microbiology laboratories.

Regardless of the method/methods that are chosen, the following aspects should be taken into consideration: (1) The method chosen has to have high sensitivity and specificity, as the probability of having a positive result is low; (2) The fact that reports of CP bacteria in companion animals are still scarce does not make the inclusion of this method unnecessary; on the contrary, as their prevalence may be underestimated and misdiagnosis of CP bacteria may have important animal and public-health consequences; and (3) in the presence of a positive result, microbiologists should follow-up the positive isolate in a specialized laboratory and advise clinicians to implement infection control procedures.

Finally, surveillance on CP bacteria in companion animals, is key to add knowledge and aid in predicting the interplay of the human–environment–animal triad on the increase in carbapenem resistance, as a means to fight the burden of AMR following the One Health concept.

## Figures and Tables

**Figure 1 antibiotics-11-00533-f001:**
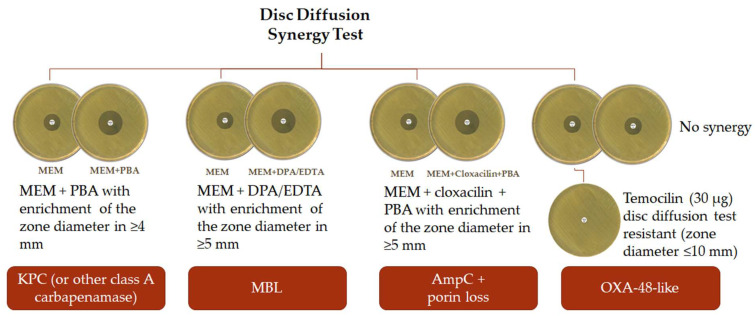
Interpretation of phenyl boronic acid (PBA), dipicolinic acid (DPA), and cloxacillin synergy tests and temocillin disc diffusion in comparation with meropenem (MEM) disc diffusion alone.

**Figure 2 antibiotics-11-00533-f002:**
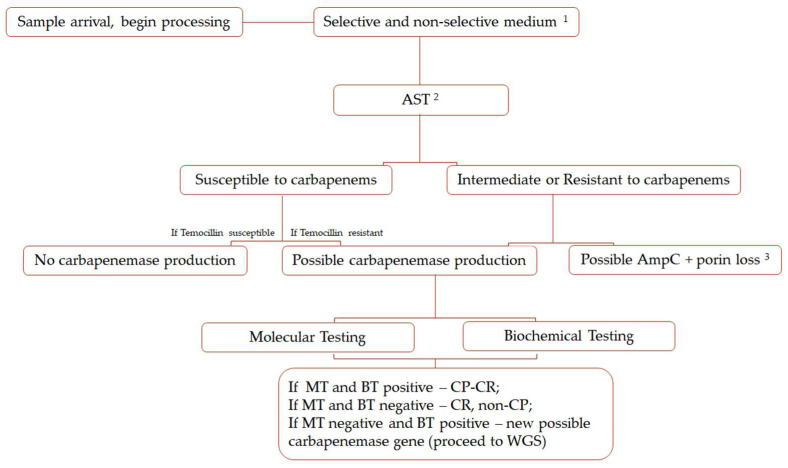
Suggested diagnostic routine for carbapenemase detection. AST, antimicrobial susceptibility testing; BT, biochemical testing; CP, carbapenemases-producing; CR, carbapenem resistant; MT, molecular testing; WGS, whole genome sequencing. ^1^ If possible, include commercially available selective media to carbapenemase-producing Enterobacterales. ^2^ AST including meropenem (10 μg) and/or imipenem (10 μg) plus temocillin (30 μg). ^3^ May be identified as described in Section 4.3.

**Table 2 antibiotics-11-00533-t002:** Common β-lactam hydrolysis profile of carbapenemases.

Amber Class	Representative Carbapenemase Type	Hydrolysis Profile				Refs.
		Narrow Spectrum Cephalosporins	Extended Spectrum Cephalosporins	Imipenem *	Meropenem *	
Class A	KPC	+	+	+	+	[2,9]
Class B	IMP, VIM, NDM,	+	+	+	+	[3]
Class D	OXA-48-like	+	-	Variable ^1^	-	[4,38,39]
	OXA-23-like	+	+	+	+	[4]

* Imipenem and meropenem representative MIC values for carbapenemase-producing isolates from companion animals are listed in Appendix A. ^1^ Imipenem susceptible in OXA-48-like has been reported.

**Table 3 antibiotics-11-00533-t003:** Characteristics of selective culture media and biochemical tests for detection of carbapenemase-producing bacteria.

Technique	Sensitivity (%)	Specificity (%)	Turnaround Time (h)	Advantages	Disadvantages
Selective Culture Medium					
SUPERCARBA	95.6–96.5	60.7	18–24	Colour identification of bacterial species	Extensive turnaround time; possible growth of non-carbapenemase producing bacteria; positive control needed.
CRE Agar	78	60–66			
ChromID CARBA Smart	91	76–89			
CHROMagar™ KPC	100	NDA			Only detects KPC-producing bacteria
CHROMagar™ OXA-48	75.8	99.3			Only detects OXA-48-producing bacteria
Biochemical Tests					
Rapidec^®^ CarbaNP	100	100	2	Rapid Detection of carbapenemase-producing bacteria	Non-specific detection; colour interpretation; expensive
CIM	NDA	NDA	8	Affordable; no commercial kit necessary	Non-specific detection; negative control strain needed; non-standardized
BlueCarba	100	100	2	Rapid Detection of carbapenemase-producing bacteria	Non-specific detection; positive control needed; expensive
β CARBA Test™	84.9	95.6	0.5	Rapid Detection of carbapenemase-producing bacteria	Non-specific detection; expensive

NDA, no data available.

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
