# Peer review of "Companion Animals—An Overlooked and Misdiagnosed Reservoir of Carbapenem Resistance"

_antibiotics, 2022, doi:10.3390/antibiotics11040533_

Round 1

Reviewer 1 Report

The most novel and relevant contribution of the review are the molecular approaches, with phenotypic methods.

 The section about the molecular detection could be extended. Nowadays work is being done on the detection of specific genes, but also on massive sequencing. I consider that the authors could do more work on this aspect, because with phenotypic methods what they describe goes beyond what is already known.

 More explanatory figures could be included because It would significantly improve their understanding

Author Response

Reviewer #1:

Reviewer#1 R1C1: The most novel and relevant contribution of the review are the molecular approaches, with phenotypic methods.
Reviewer#1 R1A1: We thank Reviewer#1 for the positive feedback about this manuscript. Bellow we have addressed the Reviewer#1 comments point by point.

Reviewer#1 R1C2: The section about the molecular detection could be extended. Nowadays work is being done on the detection of specific genes, but also on massive sequencing. I consider that the authors could do more work on this aspect, because with phenotypic methods what they describe goes beyond what is already known.
Reviewer#1 R1A2: We thank Reviewer#1 for the comment. We agree with Reviewer#1 about the relevance of the molecular methods. In the initial manuscript we did not emphasize this section to much because it is still not commonly used in veterinary diagnostic laboratories. Nevertheless, being a review manuscript, we agree that it could be improved. Therefore, changes were made, and new information was added at Page 11, starting with the following sentences: “…Similarly commercial PCR kits are also available, like the Check-MDR CT103XL (Check-Points Health, Wageningen, The Netherlands) …” and “Furthermore, the use of untargeted metagenomic next-generation sequencing (mNGS) from clinical…”

Reviewer#1 R1C3: More explanatory figures could be included because It would significantly improve their understanding.
Reviewer#1 R1A3: We thank Reviewer#1 for the comment. We gladly took your advice, and Table 3 and Figure 2 were added to the manuscript in order to summarize information regarding the techniques and make it more clear to the readers.

Reviewer 2 Report

Review of Companion Animals - an overlooked and misdiagnosed reservoir of carbapenem resistance

In this paper authors provide review of identification and detection of carbapenemase-producing (CP) bacteria in companion animals, warning of the possibility of transmission between animals and owners, veterinaries, etc .The review is up-to-date. Generally, the English language is appropriate and understandable, but since English is not my first language, I do not feel qualified to judge about language and style.

List of corrections need to be done.

  1. The authors provide an overview of the methods available for the detection of CR bacteria in clinical material in veterinary practice. However, it would be better if authors clearly give advantages, disadvantages, specificity, sensitivity and turnaround time of listed methods in table.
  2. When describing some methods, the authors mention the duration of the test and the time of obtaining the results; however this information is missing in some cases (carbapenem inactivation method, CIM) (line 243)
  3. In Figure, the authors schematically show different discs diffusion synergy tests. Why are two Petri dishes shown, each containing one antibiotic, instead of both antibiotic discs on one plate, since this is a synergy test? It should be correct.
  4. When describing tests, usually complete data on the manufacturer are given, but some are missing (line 235) it should be uniform.
  5. In chapter 4.4. Molecular Testing/ Lateral Flow Assays, authors first described lateral flow assays, than molecular testing; title suggests opposite order
  6. In the chapter Transmission potential, authors list required action to reduce health hazard. The second is “Include screening for carbapenem resistance in veterinary microbiology laboratories”. Who should be screened, which animals, in which cases?
  7. Finally, although the paper contains information on various methods used to detect CP bacteria in animals, it remains unclear what are recommendations in the routine work for veterinarians and laboratory microbiologists. Authors mentioned it in Conclusion, but it is not enough. It would be useful to describe procedures for detection CP bacteria using flowchart which include information - when to test; which animals should be tested; which methods are first choice and which are for confirmation, etc.

Suggested revisions:

Line 65:  antimicrobial - antimicrobials

Line 78: de – the

Lines 89-90: “isolated from dogs and cats clinical samples “ - clinical samples from dogs and cats

Lines 92, 93: “with the use of selective culture media being the most frequent for the detection commensal CP isolates and AST for the identification of infection CP isolates”. it is not clear and true: AST for identification? Please, revise.

Table 1: Salmonella enterica serovar Typhimurium instead of Salmonella enterica Typhimurium

Line 247: “growth is observed until the disc due to meropenem”

Line 253: “imipenem monohydrate but Tienam®…” brand name?

Line 322: “Overall, these immunochromatographic assays are a useful screening method to be used in routine microbiology when in the presence of suspected colonies” - not clear, correct it.

Line 329: “real time-PCR (RT-PCR)” –Abbreviation RT-PCR is for Reverse transcription PCR, while qPCR is for Real time PCR

Line 335: “critical hospitalized patients” – critically ill patients?

Line 386: “be reservoirs of CP bacteria to humans” – correct it

There are about six self-citations –  not sure how many are allowed.

Author Response

Reviewer #2:

Reviewer#2 R2C1: In this paper authors provide review of identification and detection of carbapenemase-producing (CP) bacteria in companion animals, warning of the possibility of transmission between animals and owners, veterinaries, etc . The review is up-to-date. Generally, the English language is appropriate and understandable, but since English is not my first language, I do not feel qualified to judge about language and style.
Reviewer#2 R2A1: We thank Reviewer#2 for the positive comments to this manuscript. Bellow we addressed all the remaining comments point by point.

Reviewer#2 R2C2: The authors provide an overview of the methods available for the detection of CR bacteria in clinical material in veterinary practice. However, it would be better if authors clearly give advantages, disadvantages, specificity, sensitivity and turnaround time of listed methods in table.
Reviewer#2 R2A2: We thank Reviewer#2 for the comment. To overcome this issue, a summarize table was added to the manuscript (Table 3, Page 8).

Reviewer#2 R2C3: When describing some methods, the authors mention the duration of the test and the time of obtaining the results; however this information is missing in some cases (carbapenem inactivation method, CIM) (line 243)
Reviewer#2 R2A3: We thank Reviewer#2 for the comment. Required, changes were made. After the sentence: “…A Mueller-Hinton Agar plate is inoculated with a known susceptible E. coli strain prior to disc placement…” the following sentence was added and re-written for better understanding: “… The turnaround time is approximately eight hours and if positive, growth is observed until the disc of meropenem due to its previous hydrolysis by the CP isolate…”

Reviewer#2 R2C4: In Figure, the authors schematically show different discs diffusion synergy tests. Why are two Petri dishes shown, each containing one antibiotic, instead of both antibiotic discs on one plate, since this is a synergy test? It should be correct.
Reviewer#2 R2A4: We thank Reviewer#2 for the question. To do the synergy test the two discs (one with carbapenem and the other with one disc that combines both drugs) may be in the same plate or in separate plates. Since the size of the image inside the article makes the visualization of the names in the disc difficult, we chose to place the two plates and the respective captions, making the visualization clearer. Furthermore, it also shows that the discs may be placed in separate plates. For this reason, we kept the image unaltered and hope that Reviewer#2 agrees.

Reviewer#2 R2C5: When describing tests, usually complete data on the manufacturer are given, but some are missing (line 235) it should be uniform.
Reviewer#2 R2A5: We thank Reviewer#2 for the suggestion, we have made changes accordingly in chapter 4.2, Page 8: “…To the best of our knowledge, the Rapidec® CarbaNP (Biomérieux, Marcy l'Etoile, France)…”.

Reviewer#2 R2C6: In chapter 4.4. Molecular Testing/ Lateral Flow Assays, authors first described lateral flow assays, than molecular testing; title suggests opposite order.
Reviewer#2 R2A6: We thank Reviewer#2 for the comment. When following suggestions from Reviewer#1 R1A2 we decided that was best to create a new section (chapter 4.5, Page 10) for Molecular testing.

Reviewer#2 R2C7: In the chapter Transmission potential, authors list required action to reduce health hazard. The second is “Include screening for carbapenem resistance in veterinary microbiology laboratories”. Who should be screened, which animals, in which cases?
Reviewer#2 R2A7: We thank Reviewer#2 for the comment and to the opportunity to explain our view on the subject. One of the purposes of this paper was to highlight the need to introduce carbapenemase detection in veterinary medicine, with a focus on companion animals. This screening should be done in all instances, regardless of species and clinical presentation due to: 1) the possibility of treatment failure; and 2) the close proximity of pets and humans. We have added a new sentence that summarizes this information in Page 13: “…The screening of CR and CP bacteria should be conducted in all companion animal samples submitted to culture and AST regardless of clinical presentation and animal species. The most reliable detection methods should be preferred according to each laboratory technical and financial availability. Figure 2 summarizes a possible workflow that can be adapted to veterinary diagnostic laboratories…”

Reviewer#2 R2C8: Finally, although the paper contains information on various methods used to detect CP bacteria in animals, it remains unclear what are recommendations in the routine work for veterinarians and laboratory microbiologists. Authors mentioned it in Conclusion, but it is not enough. It would be useful to describe procedures for detection CP bacteria using flowchart which include information - when to test; which animals should be tested; which methods are first choice and which are for confirmation, etc.
Reviewer#2 R2A8: We thank the Reviewer#2 for this kind observation. To address this comment, we have added the text referred in Reviewer#2 R2A7 and we have also created a Flowchart figure (Figure 2, Page 14) to summarize one possible workflow that with veterinary laboratories may adjust to their working routine, being applied to all samples received for microbiology analyses regardless the animals conditions.

Reviewer#2R2C9: Suggested revisions: Line 65:  antimicrobial - antimicrobials.
Reviewer#2 R2A9: We thank Reviewer#2. Changes were done accordingly.

Reviewer#2R2C10: Suggested revisions: Line 78: de – the
Reviewer#2 R2A10: We thank Reviewer#2. Changes were done accordingly.

Reviewer#2R2C11: Suggested revisions: Lines 89-90: “isolated from dogs and cats clinical samples “ - clinical samples from dogs and cats.
Reviewer#2 R2A11: We thank Reviewer#2. Changes were done accordingly.

Reviewer#2R2C12: Suggested revisions: Lines 92, 93: “with the use of selective culture media being the most frequent for the detection commensal CP isolates and AST for the identification of infection CP isolates”. it is not clear and true: AST for identification? Please, revise.
Reviewer#2 R2A12: We thank Reviewer#2. The sentence was revised.

Reviewer#2R2C13: Suggested revisions: Table 1: Salmonella enterica serovar Typhimurium instead of Salmonella enterica Typhimurium.
Reviewer#2 R2A13: We thank Reviewer#2 for noticing. Changes were done in Table 1 and in the text.

Reviewer#2R2C14: Suggested revisions: Line 247: “growth is observed until the disc due to meropenem”.
Reviewer#2 R2A14: We thank Reviewer#2. Changes were made to the sentence in order to make it clearer.

Reviewer#2R2C15: Suggested revisions: Line 253: “imipenem monohydrate but Tienam®…” brand name?
Reviewer#2 R2A15: We thank Reviewer#2 for the comment. The band name was added in Page 9: “…Tienam® (imipenem/cilastatin, Merck Sharp & Dohme, Campinas, Brazil)…”

Reviewer#2R2C16: Line 322: “Overall, these immunochromatographic assays are a useful screening method to be used in routine microbiology when in the presence of suspected colonies” - not clear, correct it.
Reviewer#2 R2A16: We thank Reviewer#2. Changes were made to clarify the information in Page 10: “…Overall, these immunochromatographic assays are useful to use as a screening method in routine microbiology when the isolated bacteria is suspected to be a carbapenemase producer…””

Reviewer#2R2C17: Suggested revisions: Line 329: “real time-PCR (RT-PCR)” –Abbreviation RT-PCR is for Reverse transcription PCR, while qPCR is for Real time PCR.
Reviewer#2 R2A17: We thank Reviewer#2 for the pertinent comment. Since real-time PCR may be used as a qualitative or quantitative method (https://www.bio-rad.com/en-pt/applications-technologies/what-real-time-pcr-qpcr?ID=LUSO4W8UU), we chose to write the full name throughout the manuscript.

Reviewer#2R2C18: Suggested revisions: Line 335: “critical hospitalized patients” – critically ill patients?
Reviewer#2 R2A18: We thank Reviewer#2. Changes were done accordingly.

Reviewer#2R2C19: Line 386: “be reservoirs of CP bacteria to humans” – correct it
Reviewer#2 R2A19: We thank Reviewer#2. Changes were done accordingly with the addition of the sentence “…be reservoirs of CP bacteria, thus acting as direct players in the transmission of these resistant bacteria to humans…”

Reviewer#2R2C20: There are about six self-citations –  not sure how many are allowed.
Reviewer#2 R2A20: We thank Reviewer#2 for the comment. We reviewed the Antibiotics publication guideline regarding self-citations. The only information we were able to find was the following: “Authors should not engage in excessive self-citation of their own work.”. We would like to argument in favor of keeping these self-citations: Our team has been working on this subject for several years which led to the description of several CP bacteria in companion animals and important longitudinal studies regarding trends in antimicrobial resistance and dissemination of bacteria between pets and humans. Being a review article, we felt that citing these papers is very relevant as they provide updated scientific data on these subjects. Since we could not find a limiting number and we consider that these citations are relevant, they were kept in the manuscript.

Reviewer 3 Report

General Comments

The manuscript provides overview of the carbapenemase detection methods that are applicable for detection of antimicrobial resistance of bacteria isolated from companion animals. There are a small number of grammar, typo and style errors. Details are provided below. The article is very well written with excellent conclusions and I recommend it for publishing in your Journal after minor revision.

Specific Comments

Abstract

  • Lines 22-23 and 26-28. Confusing sentence - check grammar

Introduction

  • Lines 47-52. Unnecessary repetition "exceptional circumstances" - this paragraph could be shorter.
  • Lines 55-58. Please provide reference for this statement.

3. Phenotypic characteristics of carbapenemases and its genetic background in isolates 101 from companion animals.

  • Line 141. Full name should be written.

4. Methods for detection and identification of carbapenemases

  • Lines 246-251. Confusing sentences check grammar.
  • Lines 332-333. Confusing sentences check grammar.

Author Response

Reviewer #3:

(General Comments):

Reviewer#3 R3C1: The manuscript provides overview of the carbapenemase detection methods that are applicable for detection of antimicrobial resistance of bacteria isolated from companion animals. There are a small number of grammar, typo and style errors. Details are provided below. The article is very well written with excellent conclusions and I recommend it for publishing in your Journal after minor revision.
Reviewer#3 R3A1: We thank Reviewer#3 very much for the positive feedback. The Specific comments were addressed bellow.

(Specific Comments):

Reviewer#3R3C2: Abstract: Lines 22-23 and 26-28. Confusing sentence - check grammar.
Reviewer#3 R3A2: We thank Reviewer#3. To clarify this issue, the two sentences were changed to: “…Like in humans, Escherichia coli and Klebsiella pneumoniae are the most represented CP Enterobacterales found in companion animals, alongside with Acinetobacter baumannii …”  and “…upmost importance to ensure an adequate monitoring and detection of CP bacteria in veterinary microbiology in order to safeguard animal health and minimise its dissemination to humans and the environment…”, respectively.

Reviewer#3R3C3: Introduction: Lines 47-52. Unnecessary repetition "exceptional circumstances" - this paragraph could be shorter.
Reviewer#3 R3A3: We thank Reviewer#3 for the pertinent comment. As suggested, in Page 2, the paragraph was shortened and the repetition deleted.

Reviewer#3R3C4: Lines 55-58. Please provide reference for this statement.
Reviewer#3 R3A4: We thank Reviewer#3 for noticing the lack of reference. A reference paper (doi: 10.1093/jac/dkw481) was added regarding this matter.

Reviewer#3R3C5: 3. Phenotypic characteristics of carbapenemases and its genetic background in isolates 101 from companion animals: Line 141. Full name should be written.
Reviewer#3 R3A5: We thank Reviewer#3. The full name for this bacterium has already been mentioned on section “2. Carbapenemase producing bacteria in companion animals”.

Reviewer#3R3C6: 4. Methods for detection and identification of carbapenemases: Lines 246-251. Confusing sentences check grammar.
Reviewer#3 R3A6: We thank Reviewer#3. As Reviewer 2 had already mentioned this paragraph, the last three lines were re-written accordingly in Page 9: “…The turnaround time is approximately eight hours and if positive, growth is observed until the disc of meropenem due to its previous hydrolysis by the CP isolate…”

Reviewer#3R3C7: 4. Methods for detection and identification of carbapenemases: Lines 332-333. Confusing sentences check grammar.
Reviewer#3 R3A7: We thank Reviewer#3. To overcome this issue, the sentence that comes after “…carbapenemase genes and colistin resistance mcr-1 gene, such as the Novodiag®CarbaR+ (Mobidiag, Espoo, Finland)…” in Page 11 was re-written accordingly: “…This Novodiag®CarbaR+ test can be applied to fresh bacterium isolates or directly from rectal swabs, having a sensitivity and specificity of 98.2% and 99.7%, respectively…”
